# Anatomically-Inspired Robotic Finger with SMA Tendon Actuation for Enhanced Biomimetic Functionality

**DOI:** 10.3390/biomimetics9030151

**Published:** 2024-03-01

**Authors:** Renke Liu, Huakai Zheng, Maroš Hliboký, Hiroki Endo, Shuyao Zhang, Yusuke Baba, Hideyuki Sawada

**Affiliations:** 1Department of Pure and Applied Physics, Waseda University, Tokyo 169-8555, Japanhendo0218@akane.waseda.jp (H.E.);; 2Department of Cybernetics and Artificial Intelligence, Faculty of Electrical Engineering and Informatics, Technical University of Košice, Letná 9, 040-01 Košice, Slovakia; maros.hliboky@tuke.sk; 3Faculty of Science and Engineering, Waseda University, Tokyo 169-8555, Japan; sawada@waseda.jp

**Keywords:** biomimetic robotics, anatomic systems, shape-memory alloys, generalist robots

## Abstract

This research introduces an advanced robotic finger designed for future generalist robots, closely mimicking the natural structure of the human finger. The incorporation of rarely discussed anatomical structures, including tendon sheath, ligaments, and palmar plates, combined with the usage of anatomically proven 3D models of the finger, give rise to the highly accurate replication of human-like soft mechanical fingers. Benefiting from the accurate anatomy of muscle insertions with the utilization of Shape Memory Alloy (SMA) wires’ muscle-like actuation properties, the bonding in-between the flexor tendons and extensor tendons allows for the realization of the central and lateral band of the finger anatomy. Evaluated using the computer vision method, the proposed robotic finger demonstrates a range of motion (ROM) equivalent to 113%, 87% and 88% of the human dynamic ROM for the DIP, PIP and MCP joints, respectively. The proposed finger possesses a soft nature when relaxed and becomes firm when activated, pioneering a new approach in biomimetic robot design and offering a unique contribution to the future of generalist humanoid robots.

## 1. Introduction

In the vast expanse of human anatomy, hands stand out as marvels of evolution, embodying unmatched versatility and dexterity. Human hands are driven by muscles located in the fingers, palms, and arms, which provide a vast number of possible output combinations and enable highly controlled motion, giving humans the ability to interact with their environment and each other freely.

Unlike the human hand’s intricate assembly of muscles and bones, traditional robotic hands, though prevalent in manufacturing, offer a stark contrast with their limited flexibility and reliance on hard materials. However, for tasks that need complex motion and require interaction with non-standard shapes, manual interventions are still necessary. The versatility of the human hand provides a significant advantage in these scenarios. However, achieving this versatility is challenging. This discrepancy underscores the need for advancements in robotic hands, particularly for tasks requiring complex, nuanced movements.

Over recent decades, the pursuit of biomimetic hands has yielded remarkable progress, embracing diverse actuation methods like pneumatic, motor-driven, and Shape Memory Alloy (SMA) technologies.

Recently, soft biomimetic hands have garnered considerable attention. For example, flexible materials such as Thermoplastic Polyurethane (TPU) and Polydimethylsiloxane (PDMS) are commonly used for the main body to replace entire hands [1] or individual fingers [2,3] in all-soft robots. Shape Memory Alloy (SMA) wires are inserted into these materials to drive the extension and flexion motions by utilizing their contraction properties. Additionally, by reducing the amount of material used at the joints in a robotic finger [4,5] or by employing different materials [6], the extension and flexion motions of human fingers can be more smoothly and anthropomorphically simulated. Owing to their incorporation of pliable materials, all-soft robotic hands excel at manipulating items that are either fragile or small. Therefore, soft robotic hands offer advantages such as safe motion, quiet operation, and light weight. 

Considering semi-soft robotic hands, especially in the research of biomimetic robotic hands, the conventional approach to design an anthropomorphic robotic hand usually uses mechanical components to replicate biological structure, typically using universal joints to replicate finger joints [7,8], using concise geometric shapes to replace the complex forms of human finger bones [6], and using SMA wires [9] or cable [10] to replicate tendons for driving robotic fingers. 

In the preceding study, tendon-driven actuators, tailored specifically for use in robotic and wearable devices were introduced [11]. Such an antagonistic SMA wire-driven robotic finger effectively achieved electrical insulation for the SMA wire while maintaining its responding speed without a significant reduction. In the following work, they introduced a novel cooling design for SMA-driven tendon actuators, aimed at providing better responsiveness for biomimetic applications [12]. Experiments with robotic joint prototypes show the concept’s success, enhancing responsiveness and stability within 0.5 s. However, only modular components are utilized to construct basic finger structures. Simple shafts are used for joints without considering the complexity of human finger anatomy, such as the shape of the bones, and the structure of the joints. Approaches like this use traditional mechanical structures for replicating anatomy structures, resulting in the loss of some of the physiological advantages, and the absence of a necessary 1-DOF for opening and closing of the palm.

The phalanx is usually considered the basic geometry in most research; however, it has a more irregular shape in the human hand. One important reason for this is to precisely distribute the pressure received by the joint. Research suggests that a 1 kg force applied to a tip can lead to 12 kg of compression at the CMC joint (joint of thumb and hand). When considering a power grip, this compression might leap to approximately 120 kg [13].

Existing widely deployed robot systems are typically bulky and cumbersome, and are primarily suited for heavy and massive industrial labor, or for repetitive missions within pre-designed locations and constraints. When considering lightweight and versatile daily routines, a robot as agile and versatile as humans can possibly outperform conventional systems. Existing robots typically utilize hard materials and hinge joints, driven by motors; some research explores the use of soft silicone for the entire finger. Both approaches have limitations: being too hard or too soft makes them less similar and less adaptive compared to a human hand.

Following such a concept, a potential solution could be a robot that fully replicates the anatomical structure of a human hand. However, fully replicating the physiological structure of a human finger presents significant challenges. Given the current limitations in sourcing materials that can precisely replicate or mimic human anatomical constructs, coupled with the inherent complexities of the human body, this research aims at preserving more human biomechanical advantages.

In this research, we propose a robot finger that recreates anatomical structures using SMA wires to simulate human muscles while leveraging commonly accessible materials. We aim to replicate the structure of the human finger, focusing on components crucial to motion that are rarely mentioned in robotic studies, such as joint capsules and cartilage. The replication of muscles will also consider the actual tendon insertion points to create biomimetic motion. SMA wires are chosen for their ability to contract silently, along with their lightweight and high energy density attributes. Additionally, we employ a CV-based data tracking system to analyze the performance of the robotic finger in comparison to human data.

## 2. Human Finger Physiology

In this section, the anatomical structure and kinematics of the human fingers will be discussed, focusing on the major physiological components that are closely related to finger motion that will contribute to the biomimetic design.

### 2.1. Anatomical Structure of Human Finger

In the human hand, each finger (excluding the thumb) consists of three parts: the distal phalanx (DP), intermediate phalanx (IP), and proximal phalanx (PP). The PP connects to the corresponding metacarpal bone. The joint between the PP and the metacarpal is termed the Metacarpophalangeal (MCP) joint. Moving distally from the MCP joint, the other joints are the Proximal Interphalangeal (PIP) joint and the Distal Interphalangeal (DIP) joint, as shown in Figure 1.

Consider the finger joint structure, a layer of articular cartilage covers the ends of the phalanx and metacarpals. The thickness of the finger cartilage is usually less than 1 mm [14]. It is primarily composed of glycosaminoglycans, proteoglycans, collagen fibers and elastin with a white, smooth, and glossy surface. Its primary function is to provide a low friction-bearing surface within the joint, it allows bones to glide over each other smoothly [15]. Due to its elastic nature, the cartilage also absorbs impacts. The entire joint is enclosed in a fibrous tissue capsule, with a synovial membrane lining its inner surface. This membrane secretes synovia, which fills the synovial cavity, providing cushioning and lubrication for the joint [16]. The synovia also supplies nutrients to the articular cartilage. 

The fibrous tissue capsule plays a role in limiting the Range of Motion (ROM). In addition to the fibrous tissue layer, ligaments and palmar plates also help maintain joint stability [17]. These structures are typically located outside the joint capsule, adjacent to or partially integrated with the fibrous tissue capsule. As shown in Figure 2, collateral ligaments, found on each side of the finger joints, prevent excessive lateral movement of the joint, enhancing its stability. Similarly, accessory ligaments serve the same purpose. At the MCP joint, they also provide stability for abduction and adduction motions [18]. Furthermore, the accessory ligaments connect to the palmar plates, keeping them in a suspended state [19]. The palmar plate is a thin rectangular sheet of fibrous thickening, comprising a proximal membranous portion and a distal fibrous portion. It connects laterally to the accessory ligaments [20], maintaining joint stability by limiting extension, and facilitates smoother movement of the flexor tendons. Figure 2b illustrates this structure.

Tendons are essential for finger movement, serving as tough connective tissue bands that link muscles to bones. A tendon is usually driven by its attached muscles, transmitting contractile force to the bone [21]. As shown in Figure 3, there are two groups of tendons in the human hand: those that flex the fingers, known as flexor tendons, and those that extend the fingers, known as extensor tendons. The flexor tendons are primarily comprised of superficial and deep flexor tendons, which ultimately insert at the base of the PIP and DIP joints. The extensor tendons, flat tendons located on the back of the hand, split into three tendon fibers after crossing the MP (Metacarpophalangeal) joint. The central tendon fiber, merging with tendon fibers from the interosseous muscles, forms the central band, which inserts at the end of the middle phalanx. The lateral fibers, fusing with tendon fibers from the interosseous muscles to form the interosseous lateral band, are inserted at the end of the distal phalanx. Figure 3a specifically illustrates this structure.

The flexor tendons of the fingers are enveloped by the tendon sheath structure in the palm and fingers. The tendon sheath is a double-layered membrane, with the synovial tendon sheath directly wrapping the tendon and the fibrous tendon sheath located externally. The synovial tendon sheath consists of two layers of synovial membrane: the visceral layer, in direct contact with the tendon, and the parietal layer on the outside. The cavity filled with synovial fluid between these two layers facilitates smoother tendon gliding and provides nutrients to the tendon [22]. In the palmar aspect of the index finger, as shown in Figure 4, the deep and superficial flexor tendons from about the midpoint of the MP joint to the termination point of the deep flexor tendon at the distal phalanx are enclosed by the synovial sheath, with the external fibrous sheath for reinforcement [23]. 

As shown in Figure 4, the sheath contains intersecting fibers forming the pars cruciformis (abbreviated as C), which is cross-shaped, and the encircling pars annularis (abbreviated as A), which is ring-like, with the latter being thicker. Its function is to prevent the flexor tendons from bowstringing away from the phalanx during finger flexion, and as the synovial sheath slides within the annular fibrous sheath, it acts like a pulley for the flexor tendons, hence also referred to as the pulley system [24].

### 2.2. Motion of Human Finger

In this section, we will discuss aspects related to the kinematics of finger motions. Typically, the human hand can perform four types of movements including Flexion, Extension, Adduction, and Abduction, which can be further grouped into two pairs.

Flexion and Extension refer to the pair of opposite motions of bringing fingers closer to and away from the palm, respectively; while Adduction and Abduction refer to the pair of opposite motions of bringing fingers closer to and away from each other, respectively. Simply put, Flexion, Extension, Abduction, and Adduction are the motions of “bending”, “straightening”, “spreading”, and “closing” of fingers, respectively.

For the four fingers excluding the thumb, the motions based on direction include flexion, extension, adduction, and abduction. The joints of the fingers, specifically the DIP and PIP joints, allow for flexion and extension, while the MCP joint allows for flexion, extension, abduction and adduction.

In many studies [7,8], researchers typically use only the contraction of a single extensor tendon to achieve extension while many other structures are also involved in this motion [16]. Here, we will only discuss the structures that are essential for replication in this research. 

The finger extension motion can be directly achieved by the extensor tendon inserted to PP, or indirectly by the combined forces of the PIP joint’s extensor tendon and the superficial flexor tendon. For the IP, its extension is facilitated by the forces transmitted to the central band from the extensor tendon and the interosseous tendons. For the DP, mainly the forces transmitted to the lateral bands from the extensor tendon and the forces from the interosseous and lumbrical muscles contribute to its extension. Regarding flexion, it is primarily accomplished by the superficial and deep flexor tendons. The superficial flexor tendon mainly acts on the PIP joint, while the deep flexor tendon primarily acts on the DIP joint but also plays a significant role in flexing the MCP and PIP joints. If the superficial flexor tendon is not functioning properly, the deep flexor tendon can take over its role [16]. Although the force might be reduced, it can still fully flex the PIP joint. These structures related to finger flexion and extension motions maintain the balance of these actions by forming antagonistic relationships.

## 3. Prototype SMA Tendon Finger

In this section, the design and fabrication of a biomimetic finger will be introduced.

### 3.1. Supporting Structure

Fingers, supported by bones, cartilage, and joint capsules, provide significant force output while having low friction and high motion speed. To preserve such advantages, and mimic natural human motion, an anatomically certified 3D model is chosen and printed by an FDM 3D printer with PLA. The cartilage between phalanxes provides lubrication for motion while absorbing impact. Considering these benefits, robot cartilage is made to reduce sliding friction. Regarding replication, warping the cartilage in fluid as the human body does gives difficulty in sealing and introduces too much complexity, thus a simplified structure is designed. A 1–2 mm thick sheet is made and fixed by adhesive to simulate human cartilage. Resin combined with equal parts of epoxy and polyamide is chosen for its good malleability and good self-lubricating properties after curing. The sheets made are applied to the basis and caputs of the phalanx (ends of the bones), and metacarpals, then cured for 24 h. The 3D-printed bones covered with this artificial cartilage are shown in Figure 5.

The human joint capsule has two layers of membrane, composing an external fibrous tissue layer and an internal synovial membrane, which is considered challenging to replicate. To simplify, the joint capsule is reduced to a single-layer membrane, keeping only the external fibrous tissue layer. For this structure, a material that is sufficiently soft, offers good sealing and can restrict abnormal motion is considered better. Eventually, latex was chosen for replicating the joint capsule. A latex tubing with an external diameter of 6 mm and an inner diameter of 5 mm is wrapped around the ends of the bones and extended backward, with the adhesive applied to the non-stressed parts, as shown in Figure 5a. Stronger fixation by using screws and washers will be applied together with other structures.

### 3.2. Tendon Sheath and Palmar Plate for Flexible Motion

Ligaments are primarily composed of collagen and elastic fibers, granting them strength and extensibility. Due to their toughness, ligaments enhance the stability of the bones and restrict their range of motion. For replicating ligaments, an elastic fiber bundle with high stretchability and durability is ideal. Providing moderate elasticity, Kinesio tape is used. Developed by Dr. Kenso Kase in the 1970s, Kinesio tape techniques can support injured muscles and joints, and relieve pain by lifting the skin, facilitating blood and lymph flow [25]. Comprised of cotton and polyurethane, Kinesio tape can stretch to 120–140% of its original length [26].

As shown in Figure 6a, Kinesio tapes are cut into rectangular pieces to mimic the collateral and accessory ligaments. Inserting to the same points, as mentioned in Section 2.1, it connects the palmar plates. For the palmar plate, consisting of a proximal membranous portion and a distal fibrous portion, rubber was used to replicate the proximal membranous portion, fixed to the bone ends at the joint capsule (latex) with epoxy resin, and further secured with screws and washers. The distal fibrous portion, which possesses a degree of elasticity and has restorative properties was replicated with EVA foam that was cut into 1 mm thick rectangles and fixed by epoxy resin as shown in Figure 6b.

Noticing that the tendon sheaths play a key role in finger motions, in this study, we replicate the digital tendon sheaths comprising synovial and fibrous sheaths. Since replicating a synovial sheath continuously would affect finger flexion and extension, we replicated the synovial sheath at the location of the fibrous sheath. We used PTFE (Polytetrafluoroethylene) tubing with an external diameter of 1 mm and an inner diameter of 0.8 mm to replicate the synovial sheath through which the FDS tendon and FDP tendon pass. The fibrous sheath, classified as A and C types was replicated differently: the thicker A part was made using a mix of epoxy and polyamide resins, similar to the cartilage replication in Section 3.1, replicating the pars annularis of the fibrous sheath.

For the palmar plates, PTFE tubing and EVA foam were bonded with epoxy resin, and the mixture of epoxy and polyamide resins was applied over the entire palmar plate. For the phalanx, PTFE tubing was bonded and covered with epoxy resin. The pars cruciformis was replicated using Kinesio tape, cut into a cross shape and placed in the same anatomical positions. Figure 7 shows the digital tendon sheath structure of the middle phalanx.

### 3.3. SMA Tendon Driver

As a transmission system, tendons convey the torque generated by muscle contractions to each joint of the finger. In our study, we use Shape Memory Alloy (SMA) wires as actuators to replicate muscles. Initially discovered by Arne Ölander in the 1970s, SMA typically consists of multiple metals and possesses unique properties of shape memory effect and superelasticity [27]. A Ti-Ni-Cu-based SMA wire BMF-150, produced by Toki Corp. was used for the tendon driver, the same as in [12]. Having a diameter of 0.15mm, each SMA wire can contract approximately 5% in length when activated and generates significant force. Additionally, SMA wires are lightweight and flexible, offering silent activation with a significantly smaller space compared to common motors. The detailed specifications of the SMA used can be seen in [12].

Muscles including flexor digitorum superficialis (FDS), flexor digitorum profundus (FDP), extensor digitorum (ED), extensor indicis (EI), palmar interossei (PI), and dorsal interossei (DI) muscles are realized by SMA wire, their function for the second (index) finger is listed in Table 1. The maximum required tendon displacement is measured by fully deflecting the finger to the reversed direction of the muscle.

Considering the required length for the SMA wire, which contracts by approximately 5%, a wire that is at least 20 times longer than the desired contraction length is needed. To connect the SMA wire, nylon wires of 0.235 mm diameter were used. This ensured the insertion points closely resembled the actual tendons’, while also providing convenience for maintenance. For the insertion point for tendons, a 0.8 mm diameter copper pillar was used. Knots were employed to connect the SMA wires to the nylon wires, as well as to attach the nylon wires to the pillars. The specific types of knots used are detailed in Appendix A. The constructed biomimetic finger prototype is shown in Figure 8.

Figure 9 illustrates the insertion points and the routing of the muscle tendons inside the robotic finger. Cross markers indicate where the polymer tendon is bonded to the SMA wire, while dashed circles indicate the points where the polymer tendon is bonded to the finger bones. Tendons that are visible from the side view are indicated by solid curves, while tendons located on opposite sides are shown by dashed curves.

Figure 9a illustrates the insertion points of extensor muscles, including EI and ED. It also shows the tendons of the interossei muscles, including PI and DI, which mainly control the abduction and adduction motions. Figure 9b illustrates the insertion points of the flexor tendons for FDS and FDP.

### 3.4. Supporting Mechanism and Testbench

Considering the multi-wire design of the SMA-driven finger, a deflection angle can be achieved by activating only one SMA in a group. During this process, other SMA wires will have no contraction while the joint is deflected. In such cases, the non-active wires will have an extra length and cause the wire to be loosened from the bearing, resulting in poor electrical conduction. This also happens on the extensor sides.

Previous research usually places a bias spring for each SMA wire. However, the metal clips and springs used are heavy in weight and require metal manufacturing. Here, we propose the S-spring, a fully 3D-printed S-shape spring featuring light weight, small size, and simple manufacturing, as shown in Figure 10.

Figure 10b illustrates how a S-spring is applied to a tendon. When in a static state, an SMA wire retains its original length, and no active force will be output to the tendon. In this case, an extra length can be found along this wire, if another tendon is pulling the finger towards the same working direction. When the SMA wire is activated and shrinks in its length, the S-spring stretches up to its maximum length, becoming approximately a straight line. In this state, most of the tension is carried by the tendon, and the spring can slide on the tendon with certain friction, preventing potential damage from overstretching. To evaluate the effectiveness of the S-spring, the relationship between tension on the tendon and the spring displacement is discussed in an experiment measuring the displacement of the S-shape spring to the applied force, as shown in Figure 11.

A logarithmic function is used for fitting the displacement of the S-shaped spring, along the tendon direction. The estimated function is shown below:(1)$$\Delta x=0.7353\times \ln\left(f\right)-0.4625$$
where f shows the tension applied to the tendon, and Δx demonstrates the displacement along the center line.

The estimated curve is shown in Figure 11, with a coefficient of determination $R^{2}=0.9783$. When the tension applied to the tendon is small, the displacement of the S-spring quickly increases. When the applied force exceeds around 100 gF, the displacement gradually approaches its maximum displacement at a much slower speed. When the tension is over 1400 gF, the tendon on the spring is almost straight, holding most of the load. This displacement relation can be simply adjusted by adjusting the thickness of the S-spring.

To acquire a stable testing surface while keeping simplicity on adjusting, an aluminum board with punched holes is utilized for installing both the finger and SMA wires, as shown in Figure 12. Two perforated mounting plates with 1.8 mm holes arranged in an equidistant hexagonal pattern are used for mounting the prototype finger, each measuring 30 cm × 40 cm, and a thickness of 1.5 mm. The plate is secured on 20 mm size aluminum profiles with M5 screws, providing enough strength for holding all SMA wires during the experiment without observable flexing.

The prototype finger is secured using a vise, which is rigidly screwed to the aluminum profiles by four M5 screws penetrating through the perforated board. A clamping block is made to give a larger contacting area to the vise, using the same material used for making cartilage, ensuring a stable position without sliding or twisting during the experiment.

To fully activate all three finger joints, the required SMA wire length for each muscle is up to 100 cm. To simplify the discussion, the effective SMA wire lengths are organized to be approximately an integer multiple of 50 cm with a group of bearings that are secured by M2-size screws and hexagonal studs. Steel bearings of 12 mm outer diameter with 1.4 mm depth V-groove are used for keeping the SMA wire at a certain height, preventing contact with the aluminum plate. The hexagonal studs used are made of brass and nylon, secured on the surface of the aluminum punch board by firmly secured nuts. The hexagonal studs serve as electrodes at the same time through the v-groove bearings located atop the studs. All copper studs are used for public ground by connecting to the aluminum board. The resistance between the bearing and the public ground is around 0.2 Ω when the SMA wire is tensioned by the S-spring.

### 3.5. Control System for Biomimetic Motion

The robotic finger’s controlling hardware integrates with x86 architecture PCs using the FT232H chip for efficient and parallel IIC channel communication. The core of hardware design includes a custom-built PCB that integrates all the components used in the system. The PCB includes an optimized trace width, with a minimum of 1.27 mm (50 mils) and an exposed copper zone that can be supplemented with additional solder, for high-current sections of the circuit to reduce power loss and heat buildup during operation. Signal integrity is maintained through calibrated coupling capacitance in signal pairs for high-speed data transmission at a minimum of 3.4 MHz under IIC protocol between chips. Colored LEDs are put on the PCB to provide visual feedback and assist computer vision for response time assessment. Voltage requirements include 15 V for powering the SMA wires’ contraction, calculated by considering wire lengths and their resistivity with a safety margin.

A GUI is made for manually controlling the output of each channel of the PCA9685 chip, based on the Python open-source library Ttkbootstrap, as shown in Figure 13b. For each channel that is connected to the SMA wires, the output duty ratio of the circuit will be controlled based on a 1526 Hz PWM wave.

### 3.6. CV Based Analyzing System

To better understand the movements and interactions of both human and artificial fingers, we track key points on the fingers and analyze their trajectories using computer vision methods. In selecting these key points, circular features were prioritized to provide more prominent and distinct reference points. For each key point on the finger, we identify a pair of points used to calculate the angles between the individual finger segments. These key points are applied to the prototype finger as shown in Figure 12.

The points are divided into four groups, each represented by a different color, corresponding to a single bone: the DP, IP, PP, and Metacarpal bones. In each group, one marker is placed as close as possible to the center of the joint and tracked to generate a trajectory for comparison with human hands. When applying these markers, we extend the finger to a straight line by applying force to the fingertip, ensuring each joint is at a 180-degree angle. All markers are aligned in a straight line. The yellow markers on the metacarpal bones are settled 5 cm apart for camera calibration purposes. This approach enables us to quantify the motion and relationships between various finger segments, providing crucial insights into the dynamics of finger movements and their potential applications in interaction and manipulation scenarios.

A high-speed camera is used and placed 490 mm away from the central yellow point. The setup is illuminated by two photography lights, each with a color temperature of 5500 K and provides approximately 800 lumens of light. The recorded video is 1280 × 720 pixels with 480 fps and encoded into 30 fps files, resulting in a 16 times slow motion.

Beyond tracking the trajectories of key points, we also used these points to monitor the movements of individual finger segments. The resulting trajectories were subjected to a comprehensive analysis, providing detailed insights into the characteristics of finger movements. Subsequently, we compared these trajectories with those of a human finger to identify similarities and differences in their motions. This comparative analysis aims to evaluate how closely a robotic finger can mimic the movements of a human finger and to identify areas that may require further improvement. As a result, we can gain a deeper understanding of the effectiveness of robotic fingers in emulating human finger movements and identify potential areas for enhancement.

## 4. Evaluation of Biomimetic Functions

To formally evaluate the proposed SMA tendon-driven biomimetic finger, the range of motion and trajectory are tested and compared with human fingers.

### 4.1. Evaluation on Range of Motion

The range of motion (ROM) is evaluated by applying the CV-based point tracking methods described in Section 3.6, applied to both the prototype finger and several human fingers. Starting from a relaxed position, as shown in Figure 14a, the two extensor muscles are fully activated to extend the finger to the maximum position, as shown in Figure 14b. The finger is then powered off until it returns to the relaxed position. Following this, the two flexion muscles are activated, reaching maximum flexion as in Figure 14c. The finger is relaxed once more, and the cycle is repeated when it is then fully relaxed as shown in Figure 14a.

The resulting ROM is compared with human data as presented in Table 2, where the positive direction is calculated from the indicated direction. The dynamic ROM of human fingers is tested in a continuous motion over 5 s, with data calculated from [28]. Compared to the dynamic ROM of a human finger, our robot has a ROM of 113%, 87% and 88%, respectively, for the DIP, PIP and MCP joints.

Figure 15 illustrates the angles acquired from the experiment. Compared with human data, the prototype finger exhibits a smaller ROM during both flexion and extension. During the experiment, if the joints are forced into a larger angle within a human-feasible range, the system can maintain the position. This suggests high friction between the joints, which may be caused by some tendons directly sliding on top of the latex-made joint capsule.

One challenge in addressing this issue is balancing the length of the Teflon guiding tube to prevent it from obstructing the turning of the joint. For example, in a human finger, the tendon of the FDP is inserted into the tendon of the FDS and then connected to the IP. However, in our design, the tendons of FDP and FDS are routed parallel to each other with respect to the IP.

### 4.2. Evaluation of Trajectory

Using the same CV-based methods, the trajectory of the finger is analyzed. As shown in Figure 16 and Figure 17, the trajectories of the DIP, PIP joints and the fingertip are tracked relative to the MCP.

During extension, both the prototype finger and the human fingertip trajectory exhibit a “jumping point” around 5–7 cm, where the motion of the MCP joint slows down and the DIP joint experiences a sudden acceleration. As a result, the rotational center shifts from the MCP towards the DIP, causing sudden changes in the trajectory of the PIP joint and the fingertip. This phenomenon also occurs in reverse after the MCP, DIP, PIP, and fingertip align in a straight line, causing the rotational center to shift back to the MCP, around which the entire finger rotates.

The most significant difference between our robotic finger with the human finger can be observed in the trajectory of the PIP joint during flexion and extension. Analysis of the recorded slow-motion footage reveals that our robotic finger initially exhibits a large deflection of the PIP joint during the first half of flexion, followed by a deflection mostly in the MCP joint, while the PIP joint stays mostly stationary. This is similar to human finger motion; however, the PIP joint in a human finger always has a larger deflection throughout the entire flexion, eventually causing the trajectories of the fingertip and PIP joint to cross over each other.

To achieve a larger ROM, we hypothesize that precise control of the output combinations of muscles and a more accurate replication of human anatomical structures are required. For the flexion muscles of the current system, the force provided by the SMA wire is insufficient, limiting the ROM for flexion motion. Allocating muscles quantitively may help in addressing this issue.

### 4.3. Discussion

Unlike conventional servo-driven robots, the proposed robotic finger exhibits a soft nature. When the finger is in a relaxed position with no power applied, it can be bent slightly beyond its maximum dynamic ROM, similar to a human finger. The robotic finger can easily be forced into most gestures that a healthy human finger can achieve, without causing permanent damage to the system, benefiting from the bias spring, which gives the finger an outstanding robustness. The extension-flexion cycle is tested following random slapping from a human hand, and the motion shows no observable deterioration.

During the experiment, when the finger is activated towards the flexion direction, we noticed that the prototype finger has a tendency to lean towards the adduction direction, as shown in Figure 18. Initially, this was considered a defect. However, upon further investigation and experimentation, we realized that the human finger naturally exhibits this property. When making a firm fist, the second (index) to fifth (small) fingers naturally apply a larger force toward the middle finger, resulting in a strong and compact fist. This observation suggests that the leaning tendency of the robotic finger may actually be a desirable feature that mimics natural human finger movement.

Counterintuitively, when the index finger is relaxed, its phalanx bones and metacarpals are not aligned in a straight line. This misalignment persists throughout the entire flexion motion, causing the index finger to form an angle with its metacarpal bone (palm bone) rather than staying in the same plane. However, when we fully extend our hand (abduction), spreading each finger apart, the index finger aligns with its metacarpal bone. Naturally, when relaxed, the second finger exhibits a certain angle relative to its metacarpal bone.

Although the proposed system currently struggles with grasping, we see a promising future for the proposed design. The current difficulty primarily comes from the absence of fat and skin tissues, which makes the robot’s finger slippery due to the exposed fabric. Molding casting with silicone layers of different stiffness may provide a good simulation of human tissue. Another area for improvement is the proper allocation of muscle equivalents in the robotic finger. The current system simplifies the discussion by using one SMA wire for each muscle; however, for the flexion motion, the force generated by the 0.15 mm SMA wire is found to be insufficient. By referring to the correct amount of muscle in a human finger, using different numbers of wires could yield better performance. Considering the difficulty of predicting the shrinkage of SMA wires, especially over repeated activation, feedback control methods can be adopted for more precise motion, with recent advancements in deep learning methods.

Building upon this concept, a complete robotic hand can be developed in the future to validate more complicated and precise gestures and actions such as grasping and griping. The proposed design could broaden the possibilities for generalist humanoid robots.

## 5. Conclusions

In this paper, we proposed a biomimetic finger based on anatomical structures actuated by Shape Memory Alloy muscles. Our finger replicates the human phalanges of the index finger, along with cartilage, tendon sheaths, ligaments, and palmar plates. By applying a PWM signal to SMA wires, we achieved biomimetic motion. Additionally, we developed a computer vision-based point tracking system to analyze the bending angles of the biomimetic fingers and compared them with those of human fingers. The results of the angles and motion trajectories of both the human and biomimetic fingers indicate the biomimetic capabilities of the proposed finger. Through numerous experimental validations, we have confirmed the effectiveness and feasibility of the biomimetic finger. Moreover, it exhibits natural characteristic softness when relaxed and hardness when activated, similar to a human finger. Through research on the proposed device, we have also identified some commonly overlooked aspects of human kinematics, suggesting that our device holds significant potential in aiding the understanding of human structure and offering new insights into the analysis of complex human motion.

## Figures and Tables

**Figure 1 biomimetics-09-00151-f001:**
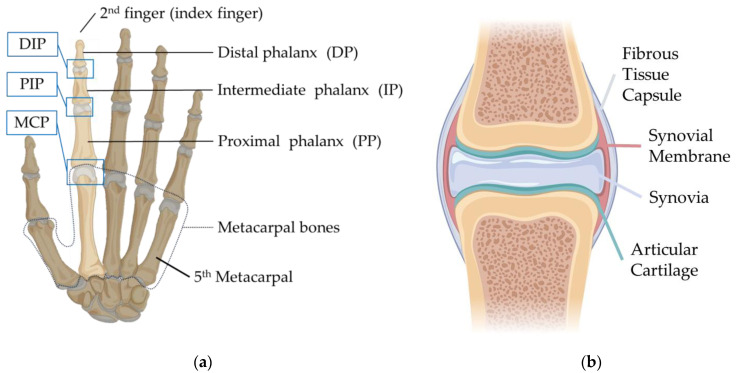
Illustration of human skeletal structures: (**a**) structure of the left-hand skeleton (palm side); (**b**) structure of the synovial joint.

**Figure 2 biomimetics-09-00151-f002:**
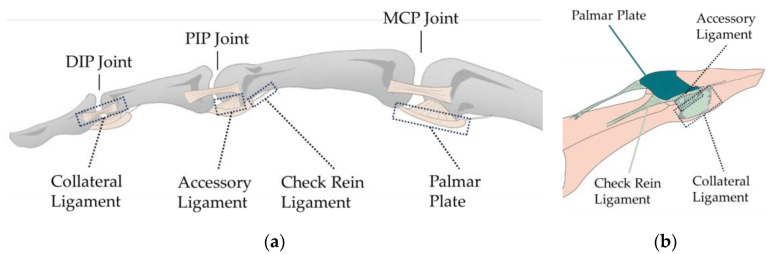
Illustration of ligaments and palmar plates: (**a**) collateral and accessory ligaments; (**b**) check rein ligament and palmar plate.

**Figure 3 biomimetics-09-00151-f003:**
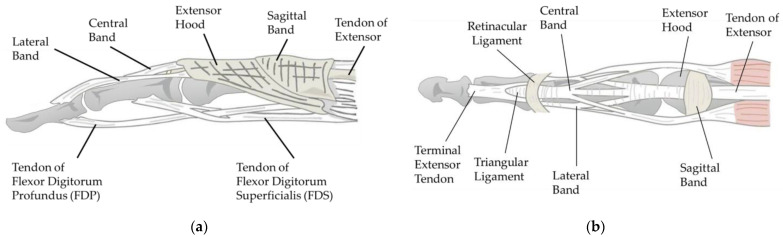
Illustrations of finger tendons: (**a**) Flexor tendon and extensor tendon lateral view; (**b**) flexor tendon and extensor tendon dorsal view.

**Figure 4 biomimetics-09-00151-f004:**
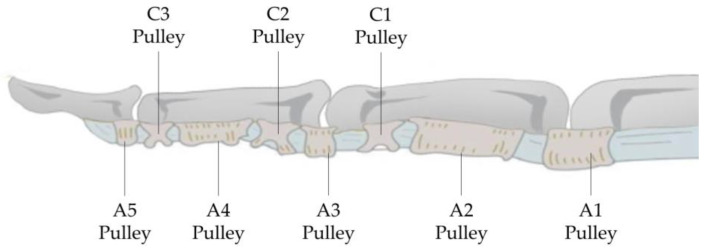
Pulley system of the Flexor Tendon.

**Figure 5 biomimetics-09-00151-f005:**
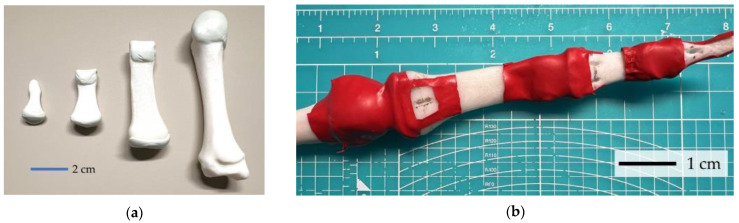
Three-dimensional printed bone with supporting structure: (**a**) finger phalanx covered with artificial cartilages made by epoxy-polyamide hybrid resin; (**b**) finger bones connected with joint capsule made by latex tube.

**Figure 6 biomimetics-09-00151-f006:**
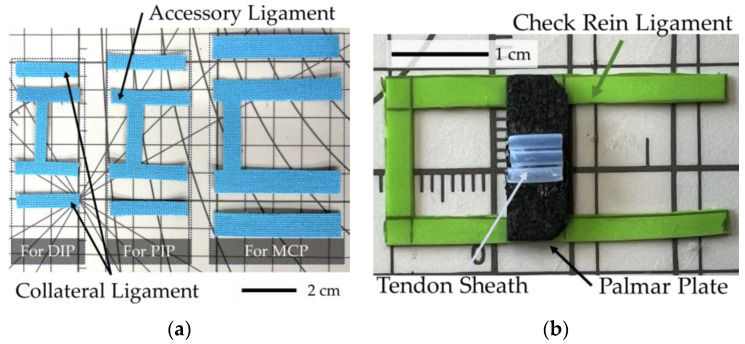
Replication of ligaments and palmar plates: (**a**) Cut Kinesio tape for replicating collateral and accessory ligament; (**b**) constructed rein ligament, palmar plates, and tendon sheath.

**Figure 7 biomimetics-09-00151-f007:**
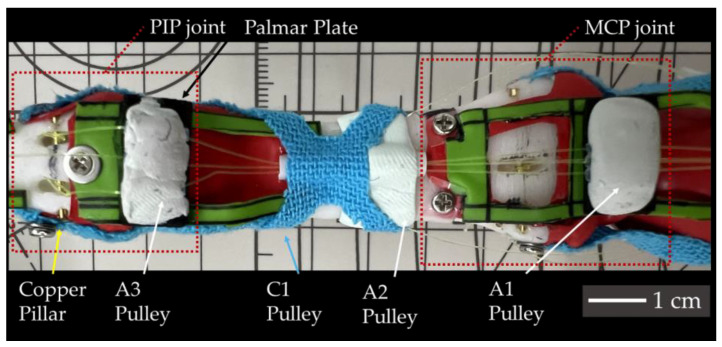
Applied structures made with Kinesio tape and resin.

**Figure 8 biomimetics-09-00151-f008:**
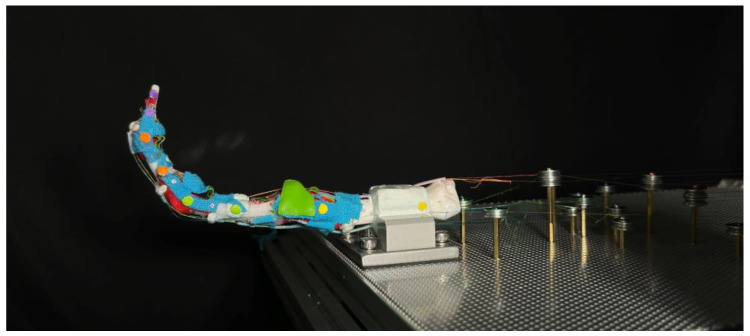
The fabricated prototype robot finger.

**Figure 9 biomimetics-09-00151-f009:**
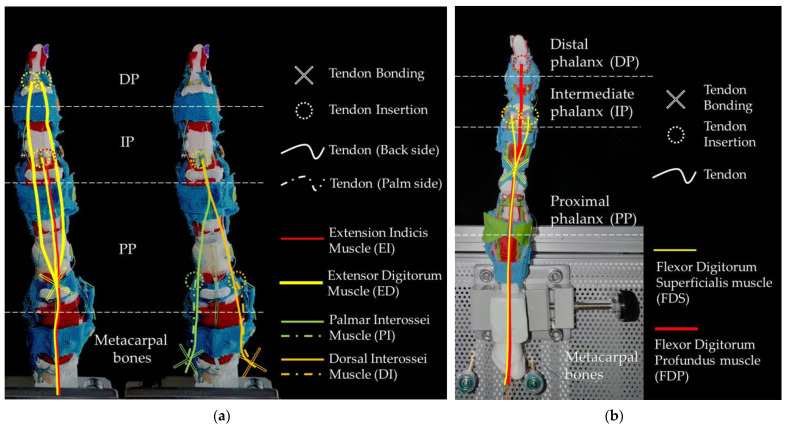
Illustration of insertion of the muscle tendons: (**a**) tendon of extension muscles (view from back side); (**b**) tendon of flexion muscles (view from palm side).

**Figure 10 biomimetics-09-00151-f010:**
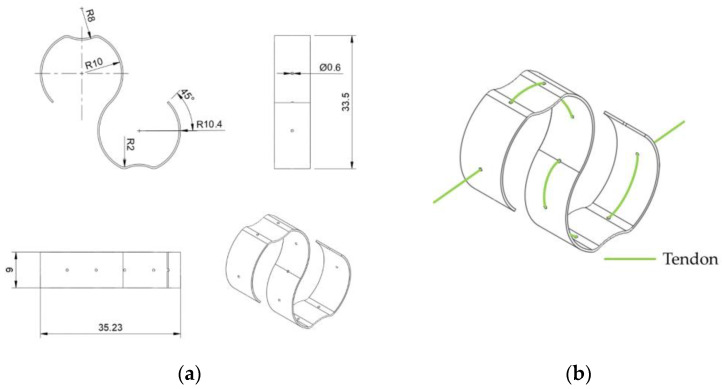
The S-shaped bias spring for SMA tendon: (**a**) dimensions of the spring (mm); (**b**) the method for applying the spring to the tendon.

**Figure 11 biomimetics-09-00151-f011:**
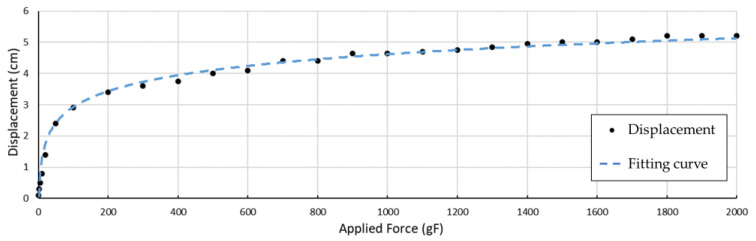
Measured spring displacement response to tension applied to the artificial tendon.

**Figure 12 biomimetics-09-00151-f012:**
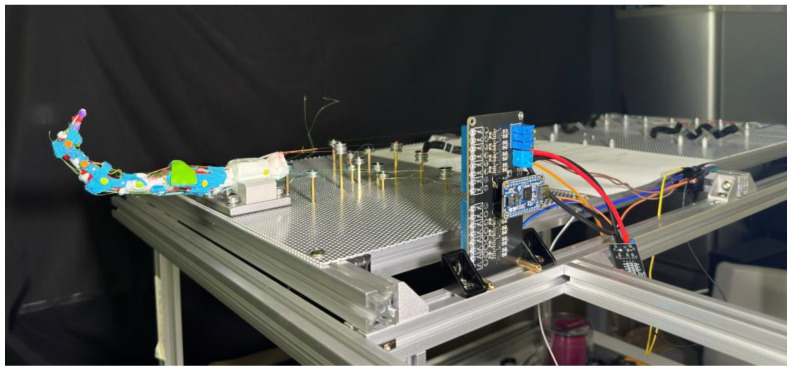
The experiment set up for prototype biomimetic finger mounted on the test bench.

**Figure 13 biomimetics-09-00151-f013:**
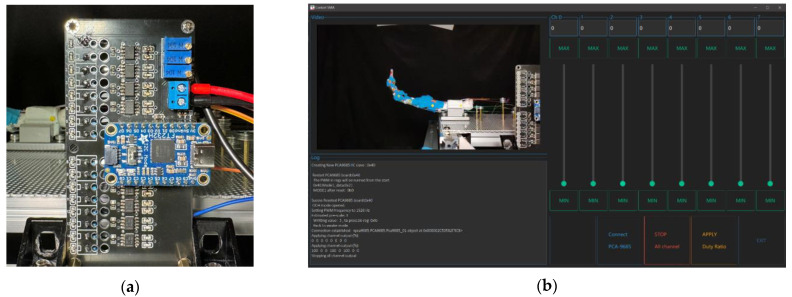
Control system for the prototype robot finger: (**a**) The printed driving circuit board; (**b**) user interface for controlling the device.

**Figure 14 biomimetics-09-00151-f014:**
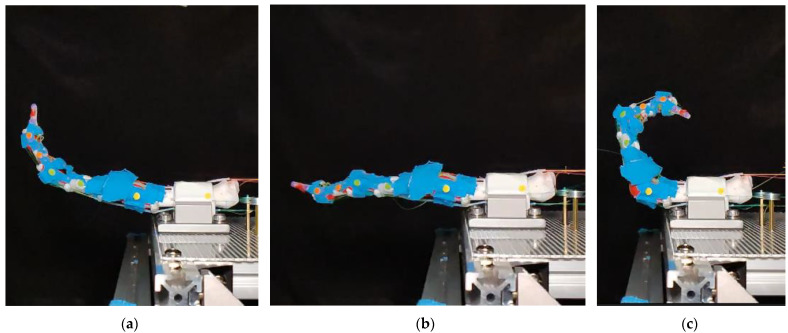
Prototype finger activated by SMA muscles: (**a**) All muscles are relaxed; (**b**) fully activated extension of SMA muscles; (**c**) fully activated flexion of SMA muscles.

**Figure 15 biomimetics-09-00151-f015:**
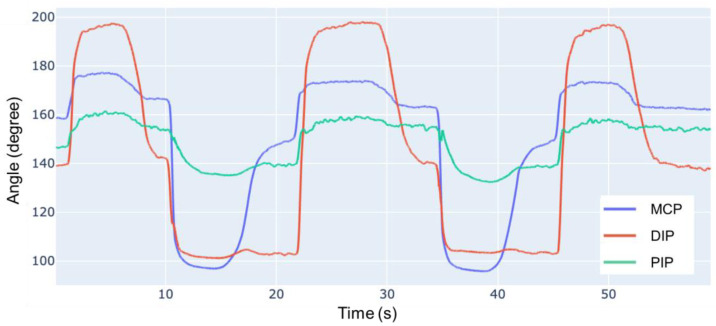
Prototype joints angles during motion.

**Figure 16 biomimetics-09-00151-f016:**
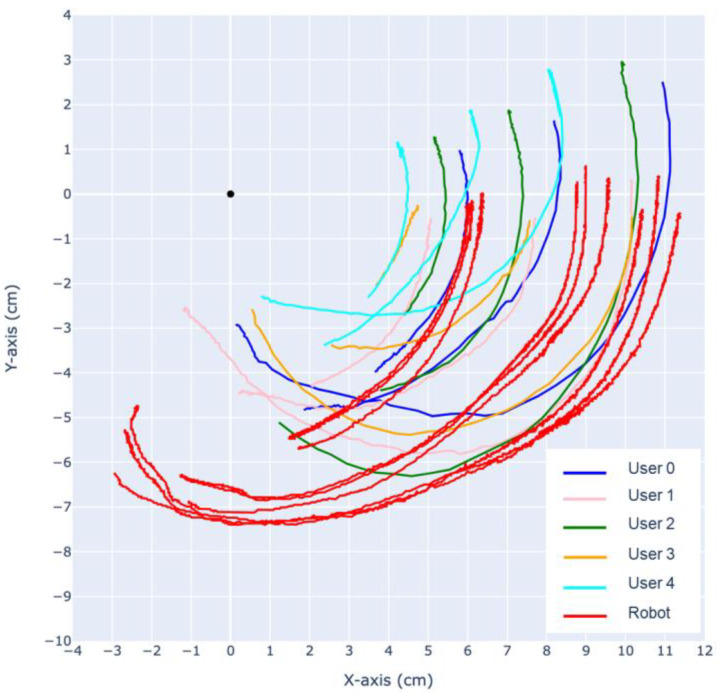
Flexion motion trajectories of human user and robot.

**Figure 17 biomimetics-09-00151-f017:**
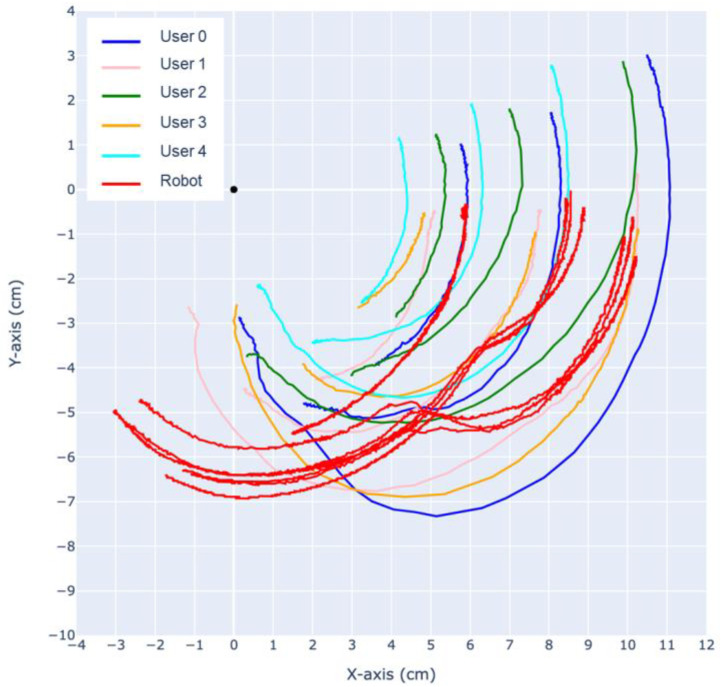
Extension motion trajectories of human user and robot.

**Figure 18 biomimetics-09-00151-f018:**
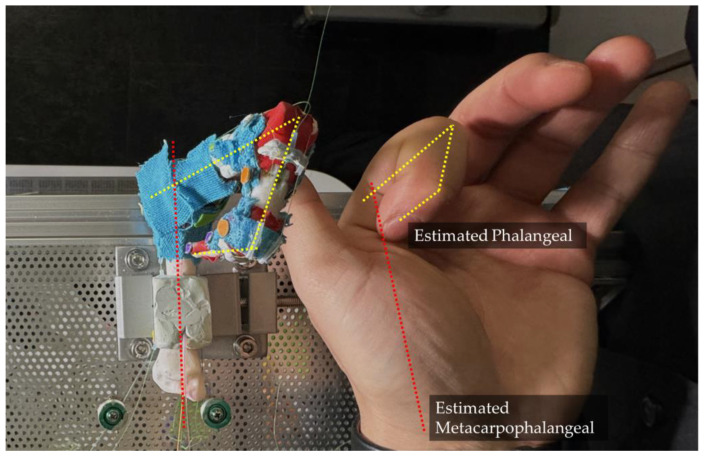
Angular deviation between the index finger and palmar plane during active flexion.

**Table 1 biomimetics-09-00151-t001:** Muscle realized by SMA wires and their function for the second (index) finger.

Motion	Muscle	Function	Required Deflection
Flexion	Flexor digitorum superficialis muscle	Flexion of PIP MCP	39 mm
	Flexor digitorum profundus muscle	Flexion of DIP PIP MCP	49 mm
Extension	Extensor digitorum muscle	Extension of DIP PIP MCP	30 mm
	Extensor indicis muscle	Extension of PIP MCP	22 mm
Adduction	Palmar interossei muscle	Adduction of MCP Flexion MCP Extension of DIP PIP	19 mm
Abduction	Dorsal interossei muscle	Abduction of MCP Extension of DIP PIP	7 mm

**Table 2 biomimetics-09-00151-t002:** Dynamic range of motion (ROM) of the prototype robot finger of each joint.

Joint	Relaxed Angle (Degree)	Type of Motion	Dynamic ROM from Relaxed Angle (Degree)	Total Dynamic ROM of Robot (Degree)	Total Dynamic ROM of Human (Degree)
DIP	31 (Flexion)	Flexion	56	82	72
		Extension	26		
PIP	40 (Flexion)	Flexion	40	101	115
		Extension	61		
MCP	22 (Flexion)	Flexion	63	83	94
		Extension	20		
	5 (Adduction)	Adduction	10 (static)	35 (static)	45 (static)
		Abduction	25 (static)		

## Data Availability

Data can be obtained from the authors upon request.

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
