# Peer review of "Anatomically-Inspired Robotic Finger with SMA Tendon Actuation for Enhanced Biomimetic Functionality"

_biomimetics, 2024, doi:10.3390/biomimetics9030151_

Round 1

Reviewer 1 Report

Comments and Suggestions for Authors

This is an interesting study. Overall, the conclusion is sound. I only have a few comments:

1, for the 2.1 introduction, many statements are missing the references. 

2, It will be better to add a scale bar in figure 5b. Just putting a ruler is not clear enough. Same issue for figure 6b and figure 7.

3, Can the authors comment on the robustness of the robotic finger? Is there any change after multiple times of movement?

4, What is the limitation and future direction?

Author Response

Thank you very much for taking the time to review this manuscript. Please find the detailed responses below and the corresponding revisions highlighted in red in the re-submitted files.

Reviewer 2 Report

Comments and Suggestions for Authors

The paper is quite interesting and deals with finger -like robotic element. In this version it is rather a project report than scientific paper, so the following improvement is needed:

- abstract and conclusions should include quantitative analysis of the results,

- the literature review should be widened with publications about materials for such constructions and methods of analysing such constructions as well as methods of studying human fingers,

- there is no info what such robot for is and though we don't know if the construction fulfilled the initial assessments of its implementation and if it is correct - please add such info,

- if the aim was to reflect the human natural movement - the study of such natural movement should be carried out - please discuss it.

Comments on the Quality of English Language

 Extensive editing of English language required

Author Response

(The authors gave the same response as above.)

Round 2

Reviewer 2 Report

Comments and Suggestions for Authors

The authors considered all my comments. Thanks! The pper, in my opinion, can be published.